# First Molecular Detection and Genetic Analysis of a Novel Porcine Circovirus (Porcine Circovirus 4) in Dogs in the World

Liu-Hui Zhang,[a] Tong-Xuan Wang,[a] Peng-Fei Fu,[b] You-Yi Zhao,[a] Hong-Xuan Li,[a] Dong-Mei Wang,[c] Shi-Jie Ma,[a] Hong-Ying Chen,[a] Lan-Lan Zheng[a]

[a]International Joint Research Center of National Animal Immunology, College of Veterinary Medicine, Henan Agricultural University, Zhengzhou, Henan Province, People's Republic of China

[b]College of Life Science and Engineering, Henan University of Urban Construction, Pingdingshan, Henan Province, People's Republic of China

[c]Lushan Dabei Agriculture and Animal Husbandry Food Co., Ltd., Lushan, Henan Province, People's Republic of China

Liu-Hui Zhang, Tong-Xuan Wang, and Peng-Fei Fu contributed equally to this work. Author order was determined based on contributions made and common consultation.

**ABSTRACT** A novel circovirus species was identified in farmed pigs and designated porcine circovirus 4 (PCV4); it has recently been proved to be pathogenic to piglets. However, little is known about its cross-species transmission, and there is no evidence of PCV4 in dogs. A total of 217 fecal samples were collected from diarrheal dogs in Henan Province, China, and tested for the presence of PCV4 using a real-time PCR assay. Among the 217 samples, the total positivity rate for PCV4 was 5.99% (13/217 samples), with rates of 7.44% and 4.17% in 2020 and 2021, respectively. PCV4 was detected in dogs in 6 of 10 cities, demonstrating that PCV4 could be detected in dogs in Henan Province, China. One PCV4 strain (HN-Dog) was sequenced in this study and shared high levels of identity (97.9% to 99.6%) with reference strains at the genome level. Phylogenetic analysis based on complete genome sequences of HN-Dog and 42 reference strains showed that the HN-Dog strain was closely related to 3 PCV4 reference strains (from pig, raccoon dog, and fox) but differed genetically from other viruses in the genus *Circovirus*. Three genotypes, i.e., PCV4a, PCV4b, and PCV4c, were confirmed by phylogenetic analysis of complete genome sequences of 42 PCV4 strains, and one amino acid variation in Rep protein (V239L) and three amino acid variations in Cap protein (N27S, R28G, and M212L) were considered conserved genotype-specific molecular markers. In conclusion, the present study is the first to report the discovery of the PCV4 genome in dogs, and the association between PCV4 infection and diarrhea warrants further study.

**IMPORTANCE** This study is the first to report the presence of PCV4 in dogs worldwide, and the first complete genome sequence was obtained from a dog affected with diarrhea. Three genotypes of PCV4 strains (PCV4a, PCV4b, and PCV4c) were determined, as supported by specific amino acid markers (V239L for open reading frame 1 [ORF1] and N27S R28G and M212L for ORF2). These findings help us understand the current status of intestinal infections in pet dogs in Henan Province, China, and also prompted us to accelerate research on the pathogenesis, epidemiology, and cross-species transmission of PCV4.

**KEYWORDS** porcine circovirus 4, dog, molecular characteristics, cross-species transmission

Porcine circoviruses (PCVs) are small, circular, single-stranded DNA viruses belonging to the genus *Circovirus* of the family *Circoviridae* (1–3). At present, there are currently four recognized types, namely, porcine circovirus 1 (PCV1), PCV2, PCV3, and PCV4. All four PCVs are similar in structure; they contain two main open reading frames (ORFs) oriented in opposite directions in the circular genome. The ORF1 or *rep* gene

Address correspondence to Lan-Lan Zheng, zhll2000@sohu.com, or Hong-Ying Chen, chhy927@163.com.

The authors declare no conflict of interest.

encodes proteins associated with replication, and the ORF2 or *cap* gene encodes the capsid or Cap protein (4, 5). Specifically, Cap is a major structural protein that contains many cell epitopes associated with viral neutralization (5).

PCV1 was first reported in 1974 and was subsequently deemed nonpathogenic to pigs (6–8), whereas PCV2 has been recognized as one of the main agents responsible for PCV-associated disease (PCVAD) (9–13). PCVAD includes postweaning multisystem wasting syndrome (PWMS), porcine dermatitis and nephrotic syndrome (PDNS), and other syndromes (9–13). PCV3 was identified by next-generation sequencing analysis in 2015, and Jiang et al. recently reported that PDNS-like disease could be reproduced in pigs infected with a cloned PCV3 virus (14–16). In 2019, a novel circovirus species was identified in farmed pigs in Hunan Province, China, and was designated PCV4 (17). Subsequently, PCV4 was reported in many provinces and cities in China and South Korea (18–22). Recently, PCV4 was successfully rescued by Niu et al. from an infectious clone and was demonstrated to be pathogenic to piglets (23).

PCV1 antibodies were detected in humans, mice, and cattle by a German group (24). PCV2 DNA can be detected in rodents, canines, ruminants, and even humans (25–29). Similar to PCV2, PCV3 DNA can be found in many animals other than pigs, such as cattle, dogs, chamois, and roe deer (3). Available data indicate that PCVs can be transmitted to nonporcine hosts, possibly via cross-species transmission routes. Cross-species transmission of PCVs is likely to be a serious threat to the global pig industry and other animal industries (30). However, few reports on PCV4 have described possible cross-species transmission events, and the status of infection in dogs remains unknown to date.

To investigate whether PCV4 DNA existed in dogs, 217 fecal samples from dogs with clinical signs of gastroenteritis (diarrhea) were collected from animal hospitals in Henan Province, China, and screened for the presence of PCV4 using a real-time PCR assay. Microbial pathogens associated with dog diarrhea were also identified, to understand the current status of intestinal infections in pet dogs in Henan Province.

## RESULTS AND DISCUSSION

**Cross-species transmission of PCV and prevalence of PCV4 in samples from dogs.** Two studies on PCV2 infection in species other than pigs showed that PCV2 might be related to reproductive failure in raccoon dogs and foxes (31, 32). PCV3 infection is associated with reproductive failure in donkeys (33). PCV4 was identified to be pathogenic to piglets by inoculation of piglets with the virus rescued from infectious clones (23). Taking lessons from PCV2 and PCV3, PCV4 is likely to be a potential threat to nonswine animals. Therefore, extensive epidemiological and etiological studies of PCV4 should be conducted in nonswine animals to better address the potential threat of this novel virus to other species.

In the present study, 217 fecal samples collected from 21 animal hospitals in Henan Province, China, in 2020 to 2021 were tested to verify the presence of PCV4. Among the 217 fecal samples, PCV4 was identified in 5.99% of the samples (13/217 samples), which was far lower than the prevalence of PCV4 (45.39% [69/152 samples]) in pigs in Henan Province described by Hou et al. (34). These differences may be attributed to different animal species and different sample types. When the data were analyzed according to year, the rates of PCV4 positivity at the sample level were 7.44% and 4.17% in 2020 and 2021, respectively. The fecal samples were collected from 10 cities in Henan Province, 6 of which were positive for PCV4. As shown in Fig. 1, the highest prevalence of PCV4 was 13.33% (2/15 samples) in Nanyang, and no positive samples were detected in Sanmenxia, Xinyang, Zhengzhou, and Anyang.

The detection results for other enteroviruses showed that the positivity rates for canine parvovirus 2 (CPV-2), canine adenovirus 1/2 (CAV-1/2), canine coronavirus (CoV), and canine distemper virus (CDV) were 69.59% (151/217 samples), 8.29% (18/217 samples), 13.82% (35/217 samples), and 6.45% (14/217 samples), respectively. Canine rotavirus (CRV) was not detected in any of the collected samples. In addition, the positivity rates for coinfections were 3.69% (8/217 samples) for CPV-2 and PCV4, 4.61% (10/217 samples) for CPV-2 and CAV-1/2, 1.38% (3/217 samples) for CPV-2 and CDV, and 1.38%

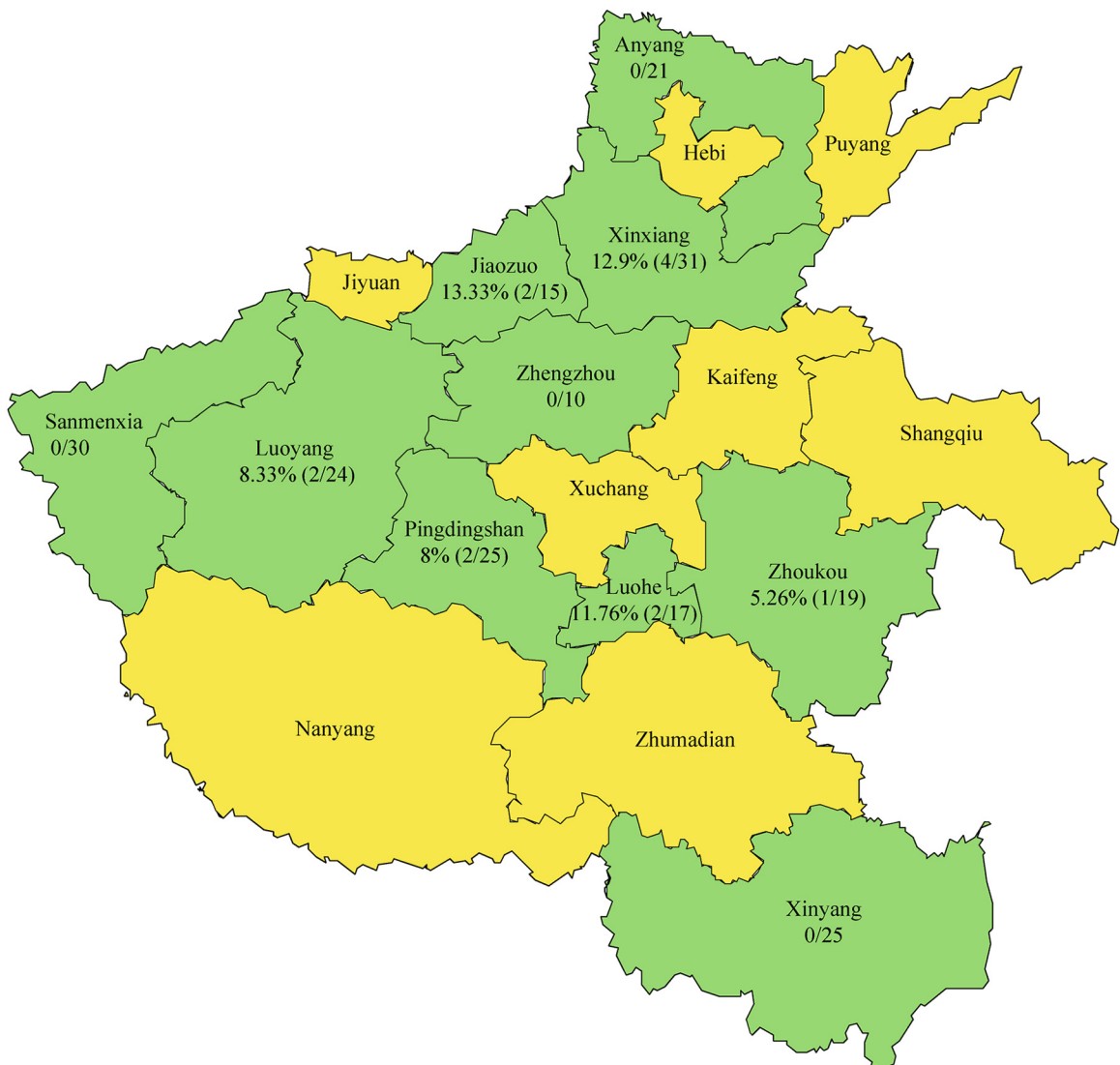

**FIG 1** Geographical distribution of the 217 samples from Henan Province, China. The numbers indicate the PCV4 positivity rates in different cities. Cities with sample collections are shaded light green, whereas cities without sample collections are shaded yellow.

(3/217 samples) for PCV4 and CAV-1/2. Interestingly, the genomes of three viruses (CPV-2, CAV-1/2, and PCV4) were detected simultaneously in one fecal sample. Only PCV4 was detected in the feces of two dogs that still had diarrhea despite deworming and antibiotics, and the other five enteroviruses were not detected. These findings suggested that CPV-2 was the main cause of diarrhea in pet dogs in Henan Province, China. According to these findings, it is likely that a single PCV4 infection or CPV-2 and PCV4 coinfection could cause diarrhea in pet dogs. Moreover, the association between PCV4 infection and diarrhea warrants further study.

Both PCV1 and PCV2 could infect human cells (35). In several studies, antibodies against PCV1 or PCV2 were detected in human serum, digestive tract, and respiratory tract samples (27, 36–39). Interestingly, PCV3 could infect nonhuman primates, but attempts to infect the human kidney 293 cell line have failed, which did not rule out infection of other human cells with PCV3 (40). PCV4, a newly discovered PCV, has been detected in only four species (pig, raccoon dog, fox, and dog) to date. Considering that dogs are human companion animals, pet dogs would be an important source of infection if PCV4 had the potential to be zoonotic. Therefore, the possible cross-species transmission of PCV4, including zoonotic transmission, warrants further investigation.

**Analysis of the homology and genetic evolution of PCV4 in dogs with respect to other species of PCV4.** To further understand the genetic characteristics of PCV4 in dogs, the complete genome of one PCV4 strain (HN-Dog) that had been collected from a dog affected with diarrhea in Luoyang, Henan Province, China, in 2021 was sequenced and deposited in the GenBank database under accession number ON937576. Similar to PCV4 determined in pigs, the complete genome of HN-Dog was 1,770 nucleotides in length, without deletions or insertions of nucleotides, and encoded two major proteins, Rep and Cap proteins, on ORFs orientated in opposite directions.

Compared with all 41 unique PCV4 strains (Table 1) available in the GenBank database (accessed 2 April 2022), the HN-Dog strain in this study showed high levels of identity (97.9% to 99.6%) at the complete genome level. Notably, of the 41 reference strains, 5 were derived from raccoon dogs (accession number MW262979 to MW262983) and 1 was derived from fox (accession number MW262984). One reference strain was from South Korea, and the other strains were from different provinces in China. In terms of cross-species transmission and transboundary aspects, high nucleotide homology among currently available PCV4 strains suggested that PCV4 had little variation. The HN-Dog strain in this study and 26 representative circovirus strains (Table 1) were selected for further analysis (Table 2). The HN-Dog strain exhibited the greatest genome identity (67.6%) with respect to mink circovirus (accession number NC_023885), followed by 62.5% with respect to bat-associated circovirus (accession number NC_038385) and 37% to 52% with respect to other circovirus species (Table 2), similar to findings reported previously (17). At the amino acid level, the identities among these circovirus strains ranged from 17% to 79.8% for Rep proteins and from 9.7% to 68.9% for Cap proteins.

**Phylogenetic analysis of PCV4 and other circoviruses.** To investigate the evolutionary relationships of PCV4 and other members of the family *Circoviridae*, a phylogenetic tree of complete genome sequences was constructed with PCV4 strain HN-Dog in the present study together with 3 PCV4 reference strains derived from three species (pig, raccoon dog, and fox) and 39 other representative circovirus strains. Phylogenetic analysis indicated that the 43 circovirus strains formed three distinct clusters (Fig. 2A). The HN-Dog strain was clustered in a large cluster with 3 PCV4 reference strains, 2 PCV1 strains, 2 PCV2 strains, and 10 other reference viruses (1 bat-associated circovirus 1, 1 bat-associated circovirus 2, 2 mink circoviruses, 2 bat circoviruses, 2 fox circoviruses, and 2 canine circoviruses). The second large cluster included 2 PCV3 strains, 1 human circovirus strain, and 2 other representative circovirus strains (1 human stool-associated circular virus and 1 *Silurus glanis* circovirus). The remaining 20 circovirus strains were located in the third large cluster. These observations were corroborated by genomic nucleotide sequence identities of the PCV4 strain in this study with reference strains (Table 2). Notably, PCV4 strain HN-Dog was clustered in an independent small branch together with 3 PCV4 reference strains (from pig, raccoon dog, and fox), indicating that they were genetically closely related.

In addition, a phylogenetic tree of complete genome sequences of 42 PCV4 strains was constructed to address the evolutionary relationships for different PCV4 strains derived from four different species, including the HN-Dog strain in this study and 41 reference strains currently available in GenBank. The phylogenetic analysis demonstrated that the 42 PCV4 strains formed three distinct clusters, namely, PCV4a, PCV4b, and PCV4c (Fig. 2B). PCV4a contained 22 PCV4 strains from four provinces (Henan, Hebei, Guangxi, and Jiangsu) in China; 6 PCV4 strains from three provinces (Fujian, Hunan, and Inner Mongolia) in China were clustered in PCV4c together with 1 South Korea strain. All strains clustered in PCV4a and PCV4c were derived from pigs. PCV4 strain HN-Dog and 13 PCV4 reference strains fell into PCV4b, with all of the strains being derived from four species (pig, dog, fox, and raccoon dog) and two adjacent Chinese provinces (Henan and Hebei). These results suggested that PCV4 could be transmitted across borders and species.

**Amino acid mutations of Cap and Rep.** Specific amino acids at position 239 of Rep and positions 27, 28, and 212 of Cap were also taken into account as proposed markers for determination of clade divisions (Fig. 2B). Concisely, PCV4a contains a combination

**TABLE 1** Information for reference strains for sequence alignment and phylogenetic analyses

| Strain | Organism | Size (bp) | Collection date | Country | Accession no. | Host |
|---|---|---|---|---|---|---|
| HNU-AHG1-2019 | PCV4 | 1,770 | February 2019 | China | MK986820.1 | Pig |
| Henan-LY1-2019 | PCV4 | 1,770 | February 2019 | China | MT015686.1 | Pig |
| KF-02-2019 | PCV4 | 1,770 | October 2019 | China | MT193105.1 | *Sus scrofa* |
| KF-01-2019 | PCV4 | 1,770 | October 2019 | China | MT193106.1 | *Sus scrofa* |
| PCV4/GX2020/NN88 | PCV4 | 1,770 | 2018 | China | MT311852.1 | *Sus scrofa* |
| PCV4/GX2020/GL69 | PCV4 | 1,770 | 2018 | China | MT311853.1 | *Sus scrofa* |
| PCV4/GX2020/FCG49 | PCV4 | 1,770 | 2018 | China | MT311854.1 | *Sus scrofa* |
| FJ-PCV4 | PCV4 | 1,770 | 2019 | China | MT721742.1 | Pig |
| JSYZ1901-2 | PCV4 | 1,770 | 2 January 2019 | China | MT769268.1 | Pig |
| E115 | PCV4 | 1,770 | 23 April 2020 | South Korea | MT882344.1 | Pig |
| PCV4/CN/NM1/2017 | PCV4 | 1,770 | 2017 | China | MT882410.1 | Pig |
| PCV4/CN/NM2/2017 | PCV4 | 1,770 | 2017 | China | MT882411.1 | Pig |
| PCV4/CN/NM3/2017 | PCV4 | 1,770 | 2017 | China | MT882412.1 | Pig |
| Hebei-AP1-2019 | PCV4 | 1,770 | 2019 | China | MW084633.1 | *Sus scrofa* |
| Hebei1 | PCV4 | 1,770 | 10 September 2020 | China | MW262973.1 | Pig |
| Hebei2 | PCV4 | 1,770 | 15 September 2020 | China | MW262974.1 | Pig |
| Hebei3 | PCV4 | 1,770 | 15 September 2020 | China | MW262975.1 | Pig |
| Hebei4 | PCV4 | 1,770 | 20 September 2020 | China | MW262976.1 | Pig |
| Hebei5 | PCV4 | 1,770 | 20 September 2020 | China | MW262977.1 | Pig |
| Hebei6 | PCV4 | 1,770 | 20 September 2020 | China | MW262978.1 | Pig |
| Hebei-Rac1 | PCV4 | 1,770 | 1 October 2015 | China | MW262979.1 | Raccoon dog |
| Hebei-Rac2 | PCV4 | 1,770 | 7 November 2017 | China | MW262980.1 | Raccoon dog |
| Hebei-Rac3 | PCV4 | 1,770 | 16 June 2019 | China | MW262981.1 | Raccoon dog |
| Hebei-Rac4 | PCV4 | 1,770 | 13 June 2018 | China | MW262982.1 | Raccoon dog |
| Hebei-Rac5 | PCV4 | 1,770 | 2 June 2018 | China | MW262983.1 | Raccoon dog |
| Hebei-Fox1 | PCV4 | 1,770 | 25 June 2018 | China | MW262984.1 | Fox |
| HN-LY-202005 | PCV4 | 1,770 | May 2020 | China | MW538943.1 | Pig |
| HN-LY-202006 | PCV4 | 1,770 | June 2020 | China | MW600947.1 | Pig |
| HN-LY-202007 | PCV4 | 1,770 | July 2020 | China | MW600948.1 | Pig |
| HN-SMX-202011 | PCV4 | 1,770 | November 2020 | China | MW600949.1 | Pig |
| HN-XX-201811 | PCV4 | 1,770 | November 2018 | China | MW600950.1 | Pig |
| HN-KF-201812 | PCV4 | 1,770 | December 2018 | China | MW600951.1 | Pig |
| HN-HB-201704 | PCV4 | 1,770 | April 2017 | China | MW600952.1 | Pig |
| HN-XX-201212 | PCV4 | 1,770 | December 2012 | China | MW600953.1 | Pig |
| HN-LY-201702 | PCV4 | 1,770 | February 2017 | China | MW600954.1 | Pig |
| HN-ZZ-201603 | PCV4 | 1,770 | March 2016 | China | MW600955.1 | Pig |
| HN-ZK-201512 | PCV4 | 1,770 | December 2015 | China | MW600956.1 | Pig |
| HN-ZK-201601 | PCV4 | 1,770 | January 2016 | China | MW600957.1 | Pig |
| HN-ZMD-201212 | PCV4 | 1,770 | December 2012 | China | MW600958.1 | Pig |
| HN-XX-201601 | PCV4 | 1,770 | January 2016 | China | MW600959.1 | Pig |
| HN-ZK-201707 | PCV4 | 1,770 | July 2017 | China | MW600960.1 | Pig |
| BaCV2 | Barbel circovirus | 1,957 | 4 May 2010 | Hungary | JF279961 | *Barbus barbus* |
| BaCV1 | Barbel circovirus | 1,957 | 2 July 2008 | Hungary | GU799606 | *Barbus barbus* |
| XOR | Bat-associated circovirus 1 | 1,862 | November 2008 | Myanmar | NC_038385 | *Rhinolophus ferrumequinum* |
| XOR7 | Bat-associated circovirus 2 | 1,798 | November 2008 | Myanmar | NC_021206 | *Rhinolophus ferrumequinum* |
| Acheng30 | Bat circovirus | 2,113 | 2016 | China | NC_035799 | *Vespertilio sinensis* |
| Daqing3 | Bat circovirus | 2,113 | 2014 | China | KX756994 | *Vespertilio sinensis* |
| FJ-FZ01 | Beak and feather disease virus | 2,003 | May 2016 | China | MG148344 | *Melopsittacus undulatus* |
| QD-CN01 | Beak and feather disease virus | 2,003 | 3 August 2008 | China | GQ386944 | *Melopsittacus undulatus* |
| CCV | Canary circovirus | 1,952 | 2003 | United Kingdom | AJ301633 | *Serinus canaria* |
| C85 | Canine circovirus | 2,063 | April 2016 | China | MK944080 | Mongrel dog |
| WM74 | Canine circovirus | 2,064 | 2015 | China | KY388502 | Dog |
| Chimp17 | Chimpanzee stool avian-like circovirus | 1,935 | September 2002 | Rwanda | GQ404851 | Chimpanzee |
| coCV | Columbid circovirus | 2,037 | 2001 | Germany | AF252610 | Pigeon |
| H51 | Swan circovirus | 1,783 | 2006 | Germany | EU056309 | Mute swan (*Cygnus olor*) |

**TABLE 1** (Continued)

| Strain | Organism | Size (bp) | Collection date | Country | Accession no. | Host |
|---|---|---|---|---|---|---|
| FJZZ302 | Duck circovirus | 1,995 | 15 December 2008 | China | GQ423747 | Duck |
| GX1104 | Duck circovirus | 1,988 | April 2011 | China | JX241046 | Duck |
| FiCV | Finch circovirus | 1,962 | 2007 | United Kingdom | DQ845075 | Finch |
| 55590 | Fox circovirus | 2,055 | 2014 | Croatia | KP941114 | *Vulpes vulpes* |
| VS7100003 | Fox circovirus | 2,063 | 3 June 2013 | United Kingdom | KP260926 | *Vulpes vulpes* |
| JX1 | Goose circovirus | 1,821 | October 2009 | Jiangxi Province, China | GU320569 | Goose |
| 24 | Gull circovirus | 2,035 | 20 August 2014 | Netherlands | KT454927 | Lesser black-backed gull |
| Unknown | Gull circovirus | 2,035 | 2009 | Germany | JQ685854 | *Chroicocephalus ridibundus* (gull) |
| VS6600022 | Human circovirus | 2,836 | 2014 | Netherlands | KJ206566 | *Homo sapiens* |
| NG13 | Human stool-associated circular virus | 1,699 | 2007 | Nigeria | NC_038392 | *Homo sapiens* |
| MiCV-DL13 | Mink circovirus | 1,753 | 30 October 2013 | China | NC_023885 | Mink |
| JL28 | Mink circovirus | 1,753 | October 2015 | China | MG001457 | Mink |
| DuCV | Mulard duck circovirus | 1,996 | 2003 | Germany | AY228555 | Duck |
| PCV1_LV34 | PCV1 | 1,759 | 2016 | Brazil | MN508363 | Swine |
| PK | PCV1 | 1,759 | 2006 | China | DQ650650 | Pig |
| LG | PCV2 | 1,768 | 11 May 2008 | China | HM038034 | Pig |
| TJ | PCV2 | 1,767 | 2009 | China | AY181946 | Pig |
| FJ-PM01/2018 | PCV3 | 2,000 | September 2018 | China | MK454951 | Pig |
| TJ-1701 | PCV3 | 2,000 | January 2017 | China | MH522791 | Swine |
| 4-1131 | Raven circovirus | 1,898 | 2005 | Australia | DQ146997 | *Corvus coronoides* |
| Bat CV | *Rhinolophus ferrumequinum* circovirus 1 | 1,760 | February 2011 | China | JQ814849.1 | *Rhinolophus ferrumequinum* |
| H5 | *Silurus glanis* circovirus | 1,966 | 26 September 2011 | Hungary | JQ011377.1 | *Silurus glanis* |
| AM-C | Starling circovirus | 2,064 | October 2012 | New Zealand | KC846095 | *Amphibola crenata* |
| Unknown | Starling circovirus | 2,063 | 2005 | Germany | DQ172906 | European starling (*Sturnus vulgaris*) |
| 32469 | Zebra finch circovirus | 1,983 | 2014 | Germany | KU641384 | *Taeniopygia guttata* (zebra finch) |

of 239V for Rep protein and 27S, 28R, and 212L for Cap protein, PCV4b contains 239L for Rep protein and 27S, 28G, and 212L for Cap protein, and PCV4c contains 239V for Rep protein and 27N, 28R, and 212M for Cap protein. In fact, amino acid substitutions as markers for clade divisions have been reported for other viruses, such as PCV3 and CPV (41–45). As sequences were added, the evolutionary tree became richer than those in previous studies (22, 34). In order to establish more accurate and scientific classification schemes, it is necessary to make greater efforts to increase the sharing of correctly annotated sequences in free databases.

The amino acid alignment of 42 PCV4 strains showed that there were 33 and 31 amino acid mutations in Rep and Cap, respectively (Fig. 3). For Rep, the N-terminal endonuclease domain containing three conserved motifs (motif I [13FTLNN17], motif II [50PHLQG54], and motif III [90YCSK93]) and the helicase domain of superfamily 3 (SF3) containing three Walker motifs (Walker A [168GxxxxGKS175], Walker B [207DDY209], and Walker C [245ITSN248]) were reported for PCV4 strains derived from pigs (46) and were also observed in PCV4 strains derived from three other species (dogs, raccoon dog, and fox).

For Cap proteins of the other three PCVs (PCV1 to PCV3), the nuclear localization signals (NLSs) that mediate nuclear targeting of viral genomes were arginine-rich regions and were experimentally confirmed (47–49); they were also predicted in the N terminus of the putative Cap of PCV4 strains derived from pigs, ranging from 1 to 38 amino acids (46). Putative NLSs were also observed in the Cap proteins of PCV4 strains obtained from three other species (dog, raccoon dog, and fox), and two amino acid variations (N27S and R28G) in the Cap protein that were used as molecular markers for

**TABLE 2** Nucleotide identity and amino acid identity between the PCV4 strain in this study and reference strains

| Organism | Strain | Host | GenBank accession no. | Identity (%) with HN-Dog | | | | | |
| | | | | Genome (n = 3) | Rep (nucleotide) (n = 3) | Rep (amino acid) (n = 3) | Cap (nucleotide) | Cap (amino acid) |
|---|---|---|---|---|---|---|---|---|
| PCV4 | HNU-AHG1-2019 | Pig | MK986820 | 98.2 | 98.3 | 99 | 97.8 | 97.4 |
| PCV4 | Hebei-Rac1 | Raccoon dog | MW262979 | 99.5 | 99.6 | 99.7 | 99.4 | 98.3 |
| PCV4 | Hebei-Fox1 | Fox | MW262984 | 99.5 | 99.4 | 99.3 | 99.7 | 99.1 |
| Barbel circovirus | BaCV1 | *Barbus barbus* | GU799606 | 41.3 | 50.6 | 46.4 | 32.1 | 25.2 |
| Bat-associated circovirus 1 | XOR | *Rhinolophus ferrumequinum* | NC_038385 | 62.5 | 69.3 | 75 | 53.7 | 45.1 |
| Bat circovirus | Acheng30 | *Vespertilio sinensis* | NC_035799 | 45.6 | 57.5 | 52.1 | 35.2 | 24.1 |
| Beak and feather disease virus | FJ-FZ01 | *Melopsittacus undulatus* | MG148344 | 40.6 | 50.6 | 44.8 | 30.7 | 27.1 |
| Canary circovirus | CCV | *Serinus canaria* | AJ301633 | 41.8 | 54.1 | 47.9 | 37.2 | 24.7 |
| Canine circovirus | C85 | Mongrel dog | MK944080 | 47.1 | 55.7 | 50.5 | 37.9 | 22.5 |
| Chimpanzee stool avian-like circovirus | Chimp17 | Chimpanzee | GQ404851 | 42.4 | 53.8 | 46 | 35.2 | 22.3 |
| Columbid circovirus | coCV | Pigeon | AF252610 | 43 | 54.6 | 48.3 | 34.6 | 22.6 |
| Cygnus olor circovirus | H51 | Mute swan (*Cygnus olor*) | EU056309 | 43.4 | 52.9 | 48.4 | 34.2 | 26.4 |
| Duck circovirus | FJZZ302 | Duck | GQ423747 | 42 | 52.9 | 48.6 | 32.9 | 25.3 |
| Finch circovirus | FiCV | Finch | DQ845075 | 42.3 | 55.2 | 48.4 | 36.1 | 24.9 |
| Fox circovirus | 55590 | *Vulpes vulpes* | KP941114 | 47.1 | 56.1 | 51.2 | 38 | 22.9 |
| Goose circovirus | JX1 | Goose | GU320569 | 41.3 | 51.8 | 46.7 | 34.4 | 24.5 |
| Gull circovirus | 24 | Lesser black-backed gull | KT454927 | 40.8 | 52.9 | 46 | 36.3 | 26.2 |
| Human circovirus | VS6600022 | *Homo sapiens* | KJ206566 | 37 | 40.9 | 25.3 | 27 | 12.3 |
| Human stool-associated circular virus | NG13 | *Homo sapiens* | NC_038392 | 45 | 52.6 | 48 | 32.3 | 18.9 |
| Mink circovirus | MiCV-DL13 | Mink | NC_023885 | 67.6 | 72.6 | 79.8 | 61 | 68.9 |
| Mulard duck circovirus | DuCV | Duck | AY228555 | 42.5 | 53.3 | 49 | 33.7 | 25.8 |
| PCV1 | PK | Pig | DQ650650 | 51.5 | 51.9 | 51 | 50.4 | 43.9 |
| PCV2 | TJ | Pig | AY181946 | 52 | 33.6 | 17 | 27.3 | 9.7 |
| PCV3 | FJ-PM01/2018 | Pig | MK454951 | 43.2 | 53 | 47.9 | 37.6 | 24.8 |
| Raven circovirus 4 | 4-1131 | *Corvus coronoides* | DQ146997 | 42.3 | 52.3 | 46.7 | 37.1 | 22.7 |
| *Rhinolophus ferrumequinum* circovirus 1 | bat CV | *Rhinolophus ferrumequinum* | JQ814849 | 47.3 | 58.2 | 49.5 | 37.4 | 33 |
| *Silurus glanis* circovirus | H5 | *Silurus glanis* | JQ011377 | 42 | 49.9 | 48.6 | 32.8 | 22.1 |
| Starling circovirus | StCV | European starling (*Sturnus vulgaris*) | DQ172906 | 43.3 | 54 | 49.5 | 33.2 | 23.3 |
| Zebra finch circovirus | 32469 | *Taeniopygia guttata* (zebra finch) | KU641384 | 42 | 54 | 47.1 | 36.6 | 25.7 |

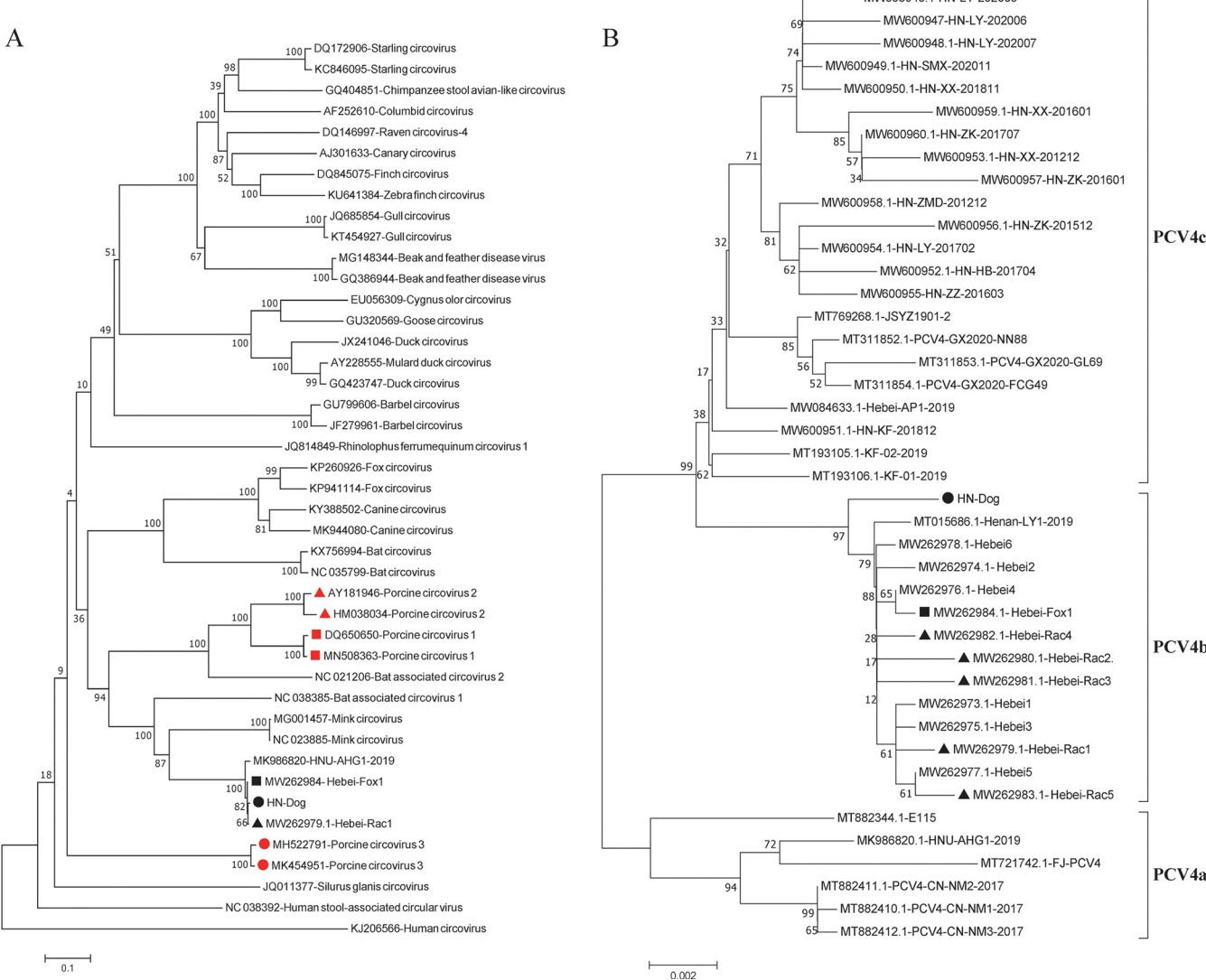

**FIG 2** NJ trees constructed with a *p*-distance model and bootstrapping at 1,000 replicates. (A) Phylogenetic tree based on the complete genomes of 43 circovirus strains, including the PCV4 strain in our study, 3 PCV4 reference strains derived from three species (pig, raccoon dog, and fox), and 39 other representative circovirus strains. All PCV4 strains cluster together independently with other representative circovirus strains. (B) Phylogenetic tree based on the complete genomes of 42 PCV4 strains. Black circles, black triangles, and black squares represent the HN-Dog strain in this study, strains from raccoon dogs, and strains from foxes, respectively. Red circles, red triangles, and red squares represent PCV1, PCV2, and PCV3, respectively. The scale bar indicates nucleotide substitutions per site.

clade division were located in the putative NLSs (Fig. 3), indicating that PCV4 strains of different genotypes might differ in cell tropism and the manner and speed of cell entry. Notably, the Cap protein of the strain in this study had one amino acid mutation (R9K), which was different from results for other PCV4 reference strains. A recent study (50) predicted five potential linear B-cell epitopes with high antigenicity, i.e., epitope A ([72]F to [88]F), epitope B ([104]N to [112]Y), epitope C ([122]D to [177]N), epitope D ([199]N to [205]V), and epitope E ([219]F to [225]P). As shown in Fig. 3, 12 amino acid substitutions were located in the predicted epitope region, and one of them (Q204H) was located in the Cap protein of the HN-Dog strain. Amino acid changes in epitope regions may be responsible for changes in the immunogenicity of Cap proteins.

**Conclusion.** Overall, this study was the first to report the presence of PCV4 in dogs in the world. The first complete genome sequence from a dog was successfully sequenced. The SCGA2022ABTC strain shared high levels of homology (97.9% to 99.6%) with other PCV4 strains. However, the pathogenicity of this virus in dogs needs to be further investigated.

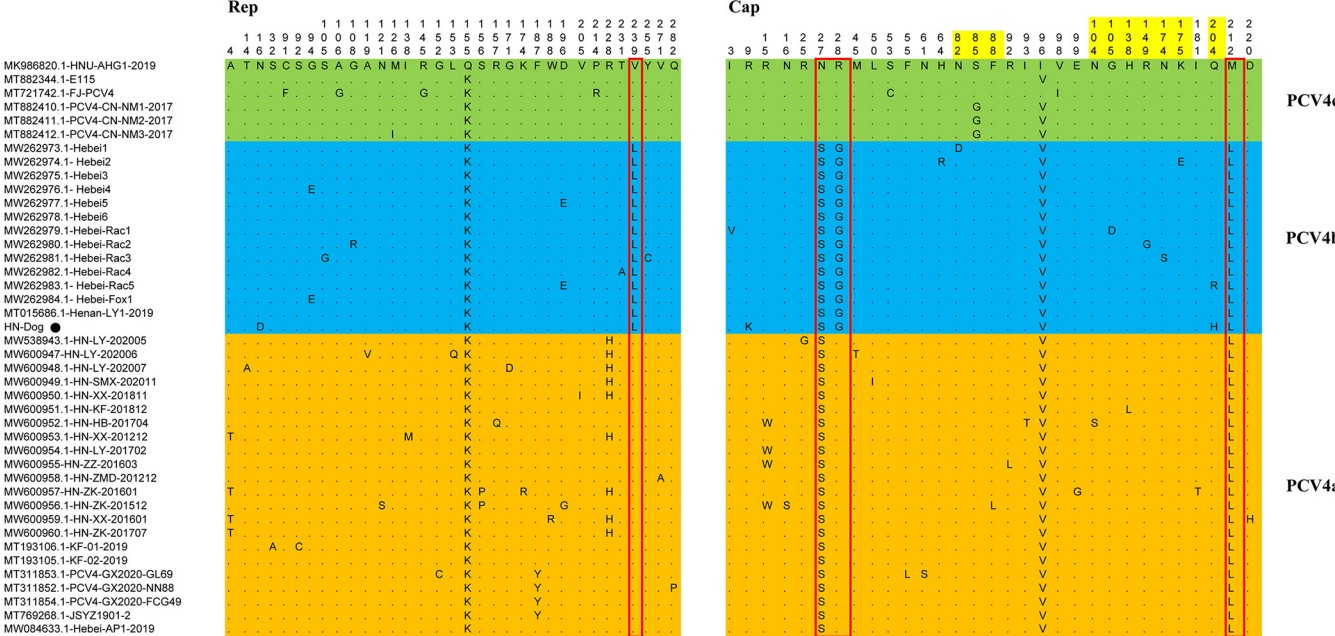

**FIG 3** All amino acid mutation sites of Rep protein and Cap protein of 44 PCV4 strains. All strains were clustered into three genotypes, namely, PCV4a (orange), PCV4b (blue), and PCV4c (light green). The potential genetic markers are shown in the red rectangles. Amino acid sites in potential epitope regions are highlighted in yellow. The black circle indicates the PCV4 strain investigated in this study.

## MATERIALS AND METHODS

**Clinical sample collection.** A total of 217 fecal samples from dogs with clinical signs of gastroenteritis (diarrhea) were collected from 21 animal hospitals located in 10 cities (Zhengzhou, Pingdingshan, Luoyang, Anyang, Xinxiang, Sanmenxia, Xinyang, Jiaozuo, Luohe, and ZhouKou) in Henan Province, China, in 2020 to 2021. After defecation, a fresh fecal sample of about 30 to 50 g (not touching the ground) was collected immediately from each dog using a sterile disposal latex glove and was placed in a disposable plastic bag. All fecal samples were stored at −80°C.

None of the animal hospitals had treated pigs, and all of the experiments were conducted in the molecular laboratory of the College of Life Science and Engineering, Henan University of Urban Construction, where no pig-related samples had been processed.

**Detection of PCV4 in clinical samples.** The fecal samples (2 g) were dissolved in an Eppendorf tube containing 10% phosphate-buffered saline (PBS) and clarified by centrifugation for 5 min at 12,000 × *g*. The supernatants were used for DNA extraction immediately or stored at −80°C until use. DNA was extracted from 200 μL of the supernatant sample using the E.Z.N.A. stool DNA kit (Omega Bio-tek, Guangzhou, China) following the manufacturer's instructions. The RNApure tissue and cell kit (Cwbio, Beijing, China) was used to extract the RNA viral genome, and then the TIANScript II reverse transcription (RT) kit (Tiangen Biotech Co., Ltd., Beijing, China) was used to acquire cDNA through RT. The DNA was screened for the presence of PCV4 using a SYBR green I-based quantitative PCR (qPCR) assay, as described previously (22). cDNA or DNA was also detected for enteroviruses in dogs, including CRV, CoV, CAV-1/2, CDV, and CPV-2, using PCR or qPCR assays, as described previously (51–55).

**Complete genome sequencing of PCV4.** To analyze the genetic diversity of PCV4, three primer pairs (Table 3) were designed to amplify three independent, overlapping DNA fragments spanning the complete genome, based on the nucleotide sequence of PCV4 (accession number MK986820.1). PCR

**TABLE 3** List of primer sequences used in this study[a]

| Primer name | Nucleotide sequence (5′ to 3′) | Primer location (nucleotide positions) | Product size (nucleotides) |
|---|---|---|---|
| PCV4-1F | GAGGTTCCACCCGTTTAAG | 260–278 | 577 |
| PCV4-1R | CCAGTCCTTGATCTGCTTGTTG | 815–836 | |
| PCV4-2F | GCCAAGACAATGTGGATTACC | 792–812 | 690 |
| PCV4-2R | AGCCTCCCATTTGCATATTACC | 1460–1481 | |
| PCV4-3F | CCACATAGTCTCCATCCAGTTG | 1361–1382 | 769 |
| PCV4-3R | CCCTCCTTTGGAGCAATACTT | 339–359 | |
| PCV4-4F | CCACATAGTCTCCATCCAGTTG | 1361–1382 | 124 |
| PCV4-4R | TACAGCCTCCCATTTGCATATTA | 1462–1484 | |

[a]Three primer pairs (PCV4-1F/R, PCV4-2F/R, and PCV4-3F/R) were used for amplification of whole-genome sequences, and PCV4-4F/R was used for detection.

was performed using a PCT-200 Peltier thermal cycler (MJ Research, Waltham, MA, USA). The PCR mixture consisted of 10 $\mu$L of PrimeSTAR Max DNA polymerase (TaKaRa, Dalian, China), 0.5 $\mu$L (25 $\mu$M) of forward and reverse primers, 1 $\mu$L of template DNA for PCV4, and 8 $\mu$L of double-distilled water. The PCR thermal conditions were as follows: initial incubation at 95℃ for 5 min, followed by 35 cycles of 95℃ for 20 s, 60℃ for 20 s, and 72℃ for 45 s. The PCR products were purified using a gel extraction kit (D2500; Omega Bio-tek) in accordance with the manufacturer's instructions. The purified products were cloned into the pMD18-T vector (TaKaRa), and the resulting recombinant plasmids were transformed into *Escherichia coli* DH-5$\alpha$ cells (TaKaRa). Three positive clones containing recombinant plasmids were independently submitted to Sangon Biotech Co., Ltd. (Shanghai, China), for sequencing by the Sanger method.

**Sequence alignment and phylogenetic analysis.** DNASTAR Lasergene and Molecular Evolutionary Genetics Analysis (MEGA) v7.0 were used for the assembly, alignment, and analysis of the sequences. A phylogenetic tree was constructed using the neighbor-joining (NJ) method in MEGA v7.0 with a *p*-distance model and a bootstrap value of 1,000 replicates.

**Ethics statement.** All experimental procedures were reviewed and approved by the Henan Agriculture University Animal Care and Use Committee (license number SCXK [Henan] 2013-0001).

**Data availability.** The PCV4 sequence obtained in our study is available from the National Center for Biotechnology Information (NCBI) (GenBank accession number ON937576).

## ACKNOWLEDGMENTS

This work was supported by the National Key Research and Development Program (grant 2021YFD1801105), the Zhongyuan High Level Talents Special Support Plan (grant 204200510015), the Henan Open Competition Mechanism to Select the Best Candidates Program (grant 211110111000), and the Program for Scientific and Technological Innovation Talents in Universities of Ministry of Education of Henan Province (grant 21HASTIT039).

Conceptualization, L.-H.Z. and H.-Y.C.; methodology, P.-F.F.; software, T.-X.W.; validation, Y.-Y.Z.; investigation, P.-F.F.; resources, H.-Y.C.; data curation, H.-X.L. and D.-M.W.; writing, original draft preparation, L.-H.Z.; writing, review and editing, L.-H.Z. and S.-J.M.; supervision, H.-Y.C. and L.-L.Z. All authors have read and agreed to the published version of the manuscript.

We declare that we have no conflicts of interest.

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
