## [Reviewer comments · Microbiology Spectrum]

Microbiology Spectrum

The first molecular detection and genetic analysis of a novel por-cine circovirus (PCV4) in dogs in the world

LiuHui Zhang, TongXuan Wang, Pengfei Fu, You-Yi Zhao, Hong-Xuan Li, DongMei Wang, ShiJie Ma, Lan-Lan Zheng, and Hong-Ying Chen

Corresponding Author(s): Hong-Ying Chen, Henan Agricultural University

Review Timeline:

Submission Date:	October 28, 2022
Editorial Decision:	December 20, 2022
Revision Received:	December 26, 2022
Accepted:	January 11, 2023

Editor: Alison Sinclair

Reviewer(s): The reviewers have opted to remain anonymous.

Transaction Report:

DOI: <https://doi.org/10.1128/spectrum.04333-22>

December 20, 2022

Prof. Hong-Ying Chen
Henan Agricultural University
河南省郑州市郑东新区河南农业大学龙子湖校区
郑州, 河南省 450002
China

Re: Spectrum04333-22 (First molecular detection and genetic analysis of the novel porcine circovirus (PCV4) in dogs with diarrhea disease)

Dear Prof. Hong-Ying Chen:

Please could you address all of the reviewer's concerns adding additional information to the manuscript where it is relevant.

Link Not Available

Sincerely,

Alison Sinclair

Journals Department
Reviewer comments:

Reviewer #1 (Comments for the Author):

Zhang, Wang, Fu and colleagues reported first detection of porcine circovirus 4 (PCV4) in diarrhea dog stool samples collected in 2020 and 2021. This study adds to the expanding literature of the detection of PCV4 in multiple animal species though the pathogenic role remains not robustly established. This study was straightforward and the manuscript was clearly written. However, both the design (no control group) and methodology (complete virus genome not prepared using proof-reading DNA polymerase) have major flaws that need to be sufficiently addressed as described below.

1. Title: Was it first detection of PCV4 in dogs globally or just in China? This should be more clearly indicated.

2. This study included sample collection from symptomatic dogs, and there was no control group, one major limitation of the current. Without an appropriate control group, it is impossible establish a casual relationship between PCV4 and diarrhea in dogs and to exclude the possibility that PCV4 detected were simply by-standers, especially co-infection rate of PCV4 with other canine diarrheal viral agents was high (85%, 11/13).
3. Thirteen dog stool samples were PCV4 DNA positive but only one complete genome was derived. Was it due to low viral load of other samples?
4. Line 243: The sybr green-based PCR method they cited can simultaneously detected both PCV2 and PCV4. As the author only sequenced complete genome of one dog PCV4 derived from the current study and presented no additional data on phylogenetic analysis of partial sequences of the rest of PCV4 PCR+ ve samples (n=12), the PCV4 specificity is not strong. This requires further elaboration or additional experimental confirmation.
5. The complete genome was sequenced by first amplifying overlapping regions using GreenTaq, followed by cloning into plasmid and then Sanger sequencing. GreenTaq appears to be not a proofreading DNA polymerase and using it for clonal sequencing was problematic, so Taq-introduced random mutation would be sequenced. This is of concern as the authors attempted to link biological meaning of mutations detected in Cap and Rep genes. At least the authors need to confirm the genome sequence using consensus amplicon sequencing prepared by high-fidelity DNA polymerase using a rolling circle method for circular DNA genome.
6. To eliminate the possibility of PCV4 detection in dogs due to environmental contamination, it would be useful to provide more information whether pigs were also treated in the animal hospitals and whether the molecular laboratory handled pig samples.
7. Line 254: What does "thermistor parameter" mean?
8. Line 262: Specify it was Sanger sequencing
9. Lines 263-267: MEGA 6 or 7 was used?
10. Phylogenetic tree: Suggest adding labels to indicate PCV types 1-4.
11. Table 3: Suggest including the purpose of the three pairs of primers. Were they for complete genome sequencing only?
12. The discussion part end abruptly with the description of rep and cap mutation. A separate, concluding paragraph should be added to sum up take home messages of the current study.

Staff Comments:

Preparing Revision Guidelines

Please return the manuscript within 60 days; if you cannot complete the modification within this time period, please contact me. If you do not wish to modify the manuscript and prefer to submit it to another journal, please notify me of your decision immediately so that the manuscript may be formally withdrawn from consideration by Microbiology Spectrum.

Manuscript Number (Spectrum04333-22R1): First molecular detection and genetic analysis of the novel porcine circovirus (PCV4) in dogs with diarrhea disease

Dear Editor and Reviewers,

Thank you for giving us the opportunity to re-submit our revised manuscript. We appreciate the concerns and suggestions provided by the reviewers and editor, and we have revised our manuscript accordingly. The modified parts in the text were indicated with red colour, and the responses to the reviewers are as following. These suggestions helped us improve the manuscript, and we hope that you find it suitable for publication.

I look forward to hearing from you soon.

Yours sincerely,

Hong-Ying Chen

Reviewers' comments:

Reviewer #1:

Comment 1: Title: Was it first detection of PCV4 in dogs globally or just in China?

This should be more clearly indicated.

Response: Thanks for your question. To our knowledge, it is the first detection of PCV4 in dogs globally. Since 2019, PCV4 has been reported in several provinces of China and Korea. However, there are no reports of PCV4 prevalence in other countries around the world. The present study was the first to report the discovery of the PCV4 genome in dogs in the world. Moreover, the title was revised according to your suggestion.

Comment 2: This study included sample collection from symptomatic dogs, and there was no control group, one major limitation of the current. Without an appropriate control group, it is impossible establish a casual relationship between PCV4 and diarrhea in dogs and to exclude the possibility that PCV4 detected were simply by-standers, especially co-infection rate of PCV4 with other canine diarrheal viral agents was high (85%, 11/13).

Response: Thanks for your question. In fact, the samples were originally used to test for diarrhea-associated viruses in dogs. Considering that PCV2 and PCV3 can be detected in dogs, we then attempted to detect PCV4 in these clinical samples. Surprisingly, PCV4 DNA was detected in these diarrhea samples. However, the prevalence of PCV4 in healthy animals was unknown and warranted further study. We also apologized for our inappropriate extrapolation that established a casual relationship between PCV4 and diarrhea in dogs, which has been revised and marked in red in the manuscript. In the future we will also try your suggestion. Samples from healthy dogs will be collected as controls for PCV4 testing.

Comment 3: Thirteen dog stool samples were PCV4 DNA positive but only one complete genome was derived. Was it due to low viral load of other samples?

Response: Thanks for your question. The reasons why only one sequence is obtained were as follows: First, the viral load of some samples was low. Second, the genome of PCV4 has a stem-loop structure that makes it difficult to amplify. Finally, four whole-genome sequences were obtained during the experiment, but they shared 100% identity. Then, they were not mentioned again in the manuscript.

Comment 4: Line 243: The sybr green-based PCR method they cited can simultaneously detected both PCV2 and PCV4. As the author only sequenced complete genome of one dog PCV4 derived from the current study and presented no additional data on phylogenetic analysis of partial sequences of the rest of PCV4 PCR+ ve samples (n=12), the PCV4 specificity is not strong. This requires further elaboration or additional experimental confirmation.

Response: We appreciate your constructive comments. In order to verify the specificity of the method, the amplification products of the detection primer pairs were submitted to Sangon Biotech Shanghai Co, Ltd., China for sequencing. Sequence alignment showed that the amplification results of the test primers belong to the PCV4 genome. The reason why only one whole genome was analyzed is shown in the “**Response to Comment 3**”.

Comment 5: The complete genome was sequenced by first amplifying overlapping regions using GreenTaq, followed by cloning into plasmid and then Sanger sequencing. GreenTaq appears to be not a proofreading DNA polymerase and using it for clonal sequencing was problematic, so Taq-introduced random mutation would be sequenced. This is of concern as the authors attempted to link biological meaning of mutations detected in Cap and Rep genes. At least the

authors need to confirm the genome sequence using consensus amplicon sequencing prepared by high-fidelity DNA polymerase using a rolling circle method for circular DNA genome.

Response: Thanks for your suggestion. According to your suggestion, PrimeSTAR® Max DNA Polymerase (Takara, Dalian, China), a high-fidelity DNA polymerase, was used to amplify the whole genome of PCV4 through three independent, overlapping DNA fragments. Sequence alignment showed that the amplification results were consistent with previous result. The most likely reason could be that three positive clones containing recombinant plasmids were independently sequenced, and no base errors occurred during sequence amplification. Thanks again for your suggestion. In the future, we will use high-fidelity enzymes for genome amplification.

Comment 6: To eliminate the possibility of PCV4 detection in dogs due to environmental contamination, it would be useful to provide more information whether pigs were also treated in the animal hospitals and whether the molecular laboratory handled pig samples.

Response: Thanks for your question. None of the animal hospitals with positive samples had treated pigs, and the molecular laboratory was specially designed to detect pathogens in pet dogs and cats. Peng-Fei Fu who contributed to the investigation in molecular laboratory of College of Life Science and Engineering, Henan University of Urban Construction was not exposed to the samples of the pigs. The purpose of these measures was to eliminate the possibility of PCV4 detection in dogs due to environmental contamination. In addition, the whole genome of HN-Dog strain was unique compared to the reference strains deposited in the GenBank database. Relevant information was added in the “MATERIALS AND METHODS” and marked with red color.

Comment 7: Line 254: What does "thermistor parameter" mean?

Response: Thanks for your question. The “thermistor parameter” implies PCR thermal conditions, which has been revised and marked with red color.

Comment 8: Line 262: Specify it was Sanger sequencing

Response: Thanks for your suggestion. The amplicons were sequenced by the Sanger method, and it has been added in the “MATERIALS AND METHODS” and marked with red color.

Comment 9: Lines 263-267: MEGA 6 or 7 was used?

Response: Sorry for our carelessness. MEGA7 was used. The corresponding content has been revised and marked with red color.

Comment 10: Phylogenetic tree: Suggest adding labels to indicate PCV types 1-4.

Response: Thanks for your suggestion. Labels denoting PCV types 1-4 have been added. In Fig. 2A, red solid circle (●), red solid triangle (▲) and red solid square (■) represent porcine circovirus 1 (PCV1), porcine circovirus 2 (PCV2) and porcine circovirus 3 (PCV3), respectively.

Comment 11: Table 3: Suggest including the purpose of the three pairs of primers.

Were they for complete genome sequencing only?

Response: Thanks for your suggestion. The three pairs of primers in Table 3 were used for complete genome sequencing only. In addition, a pair of detection primers was added in Table 3. Moreover, we annotate the purpose of primer pairs in Table 3.

Comment 12: The discussion part end abruptly with the description of rep and cap mutation. A separate, concluding paragraph should be added to sum up take home messages of the current study.

Response: Thanks for your suggestion. A separate, concluding paragraph has been added to sum up take home messages of the current study.

January 11, 2023

Prof. Hong-Ying Chen
Henan Agricultural University
河南省郑州市郑东新区河南农业大学龙子湖校区
郑州, 河南省 450002
China

Re: Spectrum04333-22R1 (The first molecular detection and genetic analysis of a novel por-cine circovirus (PCV4) in dogs in the world)

Dear Prof. Hong-Ying Chen:

thankyou for addressing the reviewers concerns.

Your manuscript has been accepted, and I am forwarding it to the ASM Journals Department for publication. You will be notified when your proofs are ready to be viewed.

Sincerely,

Alison Sinclair
Editor, Microbiology Spectrum
